# Transcript-dependent effects of the CALCA gene on the progression of post-traumatic osteoarthritis in mice
Shan Jiang [1], Weixin Xie[1], Paul Richard Knapstein [1], Antonia Donat [1], Lilly-Charlotte Albertsen[1], Jan Sevecke[1], Cordula Erdmann[1], Jessika Appelt[2], Melanie Fuchs[2], Alexander Hildebrandt[2,3], Tazio Maleitzke [2,3,4,5,6], Karl-Heinz Frosch[1,7], Anke Baranowsky[1] & Johannes Keller [1] ✉

Osteoarthritis represents a chronic degenerative joint disease with exceptional clinical relevance. Polymorphisms of the *CALCA* gene, giving rise to either a procalcitonin/calcitonin (*PCT/CT*) or a calcitonin gene-related peptide alpha (αCGRP) transcript by alternative splicing, were reported to be associated with the development of osteoarthritis. The objective of this study was to investigate the role of both *PCT/CT* and αCGRP transcripts in a mouse model of post-traumatic osteoarthritis (ptOA). WT, αCGRP$^{-/-}$ and CALCA$^{-/-}$ mice were subjected to anterior cruciate ligament transection (ACLT) to induce ptOA of the knee. Mice were sacrificed 4 and 8 weeks post-surgery, followed by micro-CT and histological evaluation. Here we show that the expression of both *PCT/CT* and αCGRP transcripts is induced in ptOA knees. CALCA$^{-/-}$ mice show increased cartilage degeneration and subchondral bone loss with elevated osteoclast numbers compared to αCGRP$^{-/-}$ and WT mice. Osteophyte formation is reduced to the same extent in CALCA$^{-/-}$ and αCGRP$^{-/-}$ mice compared to WT controls, while a reduced synovitis score is noticed exclusively in mice lacking *CALCA*. Our data show that expression of the *PCT/CT* transcript protects from the progression of ptOA, while αCGRP promotes osteophyte formation, suggesting that *CALCA*-encoded peptides may represent novel targets for the treatment of ptOA.

Osteoarthritis is a common degenerative joint disease with a global prevalence of more than 20% in individuals aged 40 and over[1]. Among other factors, joint degeneration often occurs after a joint injury, resulting in post-traumatic OA (ptOA)[2]. Affected patients experience chronic pain, restricted joint function with limited range of motion, and reduced quality of life. Although destruction of articular cartilage represents the pathological hallmark of ptOA, altered subchondral bone architecture, osteophyte formation, and synovitis are also major clinical features[3]. Previous studies[4,5] demonstrated that subchondral bone serves as a mechanic buffer for proper load perception and distribution, suggesting that progressive cartilage destruction may also result from alterations in subchondral bone architecture. Up to now, ptOA is considered incurable

without surgical intervention, resulting in exorbitant health care costs[6]. Although a number of studies contributed to our growing understanding of the molecular mechanisms underlying ptOA progression, no novel pharmacological approach has shown satisfactory outcomes regarding ptOA progression to date. For example, a recent clinical trial showed that bone morphogenetic protein 7 (BMP-7), although promoting cartilage repair, exerted similar effects on pain reduction as placebo control[7]. Another phase II trial reported that a matrix metalloproteinase inhibitor is unsuited to treat degenerative joint disease due to considerable musculoskeletal adverse effects[8]. Therefore, further studies are required to better understand the mechanisms underlying ptOA progression to identify potential treatment options.

[1]Department of Trauma and Orthopedic Surgery, University Medical Center Hamburg-Eppendorf, Hamburg, Germany. [2]Berlin Institute of Health at Charité—Universitätsmedizin Berlin, Julius Wolff Institute, Berlin, Germany. [3]Charité—Universitätsmedizin Berlin, corporate member of Freie Universität Berlin and Humboldt-Universität zu Berlin, Center for Musculoskeletal Surgery, Berlin, Germany. [4]Berlin Institute of Health at Charité—Universitätsmedizin Berlin, BIH Biomedical Innovation Academy, BIH Charité Clinician Scientist Program, Berlin, Germany. [5]Department of Orthopaedic Surgery, Copenhagen University Hospital - Amager and Hvidovre, Hvidovre, Denmark. [6]Department of Clinical Medicine, University of Copenhagen, Copenhagen, Denmark. [7]Department of Trauma Surgery, Orthopedics and Sports Traumatology, BG Hospital Hamburg, Hamburg, Germany. ✉e-mail: j.keller@uke.de

Previously it was reported that single nucleotide polymorphism in the gene encoding calcitonin (CT) is associated with the development of primary osteoarthritis of the knee[9,10]. CT and its peptide precursor, procalcitonin (PCT; hereafter referred to as *PCT/CT* transcript) are encoded by the *CALCA* gene (in humans termed *CALC1*), which also gives rise to calcitonin gene-related peptide alpha (αCGRP; hereafter referred to as *αCGRP* transcript) through alternative splicing[11]. In the healthy organism, the *PCT/CT* transcript is expressed in thyroid C cells, resulting in the release of CT into the circulation after proteolytic processing of PCT, whereas the *αCGRP* transcript is expressed in neuronal tissue[12]. While CT primarily controls bone formation by regulating the release of coupling factors from osteoclasts mediated by the CT receptor (CTR)[13], αCGRP released from sensory fibers innervating bone tissue stimulates osteoblast function through the calcitonin receptor-like receptor (CRLR)[14,15].

Despite the precise regulation of *CALCA* gene transcription, it is overexpressed in other tissues and cell types in certain medical conditions. For instance, PCT/CT is expressed ubiquitously in bacterial sepsis, where PCT negatively affects disease outcomes by signaling through the CRLR[14]. In the skeleton, our previous study[15] showed that PCT is expressed in osteoblasts during teriparatide treatment and controls bone resorption by inhibiting macrophage migration and fusion, which is required for osteoclastogenesis. Likewise, it was demonstrated that synovial fibroblasts express both the *αCGRP* mRNA transcript and mature protein, contributing to joint pain in degenerative joint disease[16].

Thus, as an association of *CALCA* polymorphism with the development of the degenerative joint disease of the knee was described previously[9,10], and as an expression of *CALCA*-encoded peptides was found to increase in joint tissue of patients with osteoarthritis[16], this study was designed to test a pathophysiologic impact of *CALCA* expression on the progression of ptOA of the knee. For this purpose, the course of ptOA following transection of the anterior cruciate ligament (ACLT) was compared in mice deficient in the *CALCA* gene with mice exclusively lacking *αCGRP* to allow a delineation of the functional impact of the *PCT/CT* and the *αCGRP* transcript. Our results show that both the *PCT/CT* and the *αCGRP* transcripts of the *CALCA* gene are induced in joint tissue following ACLT. While the *PCT/CT* transcript controls subchondral bone remodelling and protects from cartilage degeneration, the *αCGRP* transcript plays a pivotal role in osteophyte formation.

## Results

### The expression of both the *PCT/CT* and the αCGRP transcript is induced during ptOA progression

To investigate a possible role of the *CALCA* gene in ptOA progression, we first monitored the gene expression of the *PCT/CT* and *αCGRP* transcript in ACLT- and sham-operated knees of WT mice. While the *PCT/CT* transcript was overexpressed 4 weeks post-operatively and then declined, *αCGRP* expression was increased 4 and 8 weeks after ACLT (Fig. 1a). Likewise, *CALCRL*, which encodes CRLR mediating the biologic effects of both PCT and αCGRP, was induced throughout ptOA progression. Next, we studied the expression of PCT, CT, αCGRP and CRLR on protein level during the course of ACLT-induced ptOA knees in WT mice using immunofluorescent stainings. In naive knees, no signal was observed in the case of PCT and CT, whereas mild expression of CGRP and CRLR was observed primarily in the bone marrow (Supplementary Fig. 1a). At 4 weeks after ACLT, fluorescent microscopy showed that PCT and CRLR were expressed in both articular cartilage and subchondral bone marrow, while αCGRP expression was abundant in the subchondral bone in ptOA knees (Fig. 1b, c). In contrast, no specific signal for the CT protein could be detected, indicating that the induction of the *PCT/CT* transcript primarily yields the PCT protein, but not CT, during ptOA. Similar results were obtained 8 weeks after ACLT, with strong PCT and CRLR signals in articular cartilage and subchondral bone marrow, no specific CT signal in any compartment, and intense staining for CGRP in subchondral bone marrow (Supplementary Fig. 1b). Finally, we employed ELISA to measure the serum concentration of PCT, CT and αCGRP in WT mice with ptOA (Fig. 1d). No alteration was observed in the

concentrations of all three *CALCA*-encoded peptides, pointing towards a local impact of *CALCA* expression in ptOA progression of the knee.

### The inactivation of the PCT/CT transcript aggravates ACLT-induced cartilage degeneration

To further understand the role of *PCT/CT* and *αCGRP* transcripts in ptOA, *CALCA*- and *αCGRP*-deficient mice were subjected to ACLT and compared to WT controls. In *CALCA*-deficient mice, the production of PCT, CT, and αCGRP is ablated, while *αCGRP*$^{-/-}$ mice exclusively lack αCGRP protein (Supplementary Fig. 2a, b). To confirm successful inactivation of the respective *CALCA*-encoded peptides in mutant mice, we first performed immunofluorescence with PCT-, CT- and CGRP-specific antibodies 4 weeks after ACLT, at which time WT mice show strong staining intensities for both PCT and CGRP expression. While CT was undetectable in ptOA knees of mice of both genotypes, CGRP expression was absent in both *CALCA*- and *αCGRP*-deficient mice, whereas PCT expression was absent only in *CALCA*- but not *αCGRP*-deficient mice (Supplementary Fig. 2c). As cartilage degeneration is one of the major pathological changes in ptOA progression, we next investigated articular cartilage deterioration in ptOA knees of the employed mouse lines. At 4 weeks after surgery, WT ptOA knees exhibited a pronounced loss of proteoglycans, accompanied by cartilage fibrillations and erosion (Fig. 2a). Semiquantitative scoring showed that a significantly higher total OARSI score was observed in the total joint of *CALCA*-deficient mice, while the respective scores in *αCGRP*-deficient mice were significantly lower than that of either WT or *CALCA*$^{-/-}$ mice at the 4-week time point (Fig. 2b). Similarly, *CALCA*$^{-/-}$ mice showed a significantly higher score in the MFC compared to either WT or *αCGRP*-deficient mice, while a reduced OARSI score was detected in the MTP of *αCGRP*-deficient mice. At 8 weeks after surgery, no significant changes were observed in both mutant mouse lines compared to WT controls, however mice lacking *αCGRP* presented significantly lower OARSI scores compared to *CALCA*-deficient mice. To further characterize cartilage degeneration, histomorphometric analysis of the cartilage thickness was performed. In WT mice, ACLT resulted in a thinner layer of subchondral bone plate and a reduced HC/CC ratio at both 4 and 8 weeks post-operatively (Fig. 2c). For comparison of mouse lines, the SBP thickness and HC/CC ratio of ptOA knees was then normalized to those of the contralateral healthy knees and fold changes were plotted. Here, a significant reduction in SBP thickness was observed in *CALCA*-deficient mice 4 and 8 weeks after ACLT, while the HC/CC ratio was not significantly different between the three employed mouse lines at both time points studied (Fig. 2d).

### Abnormal subchondral bone architecture in *CALCA*- but not αCGRP-deficient mice

We next performed radiological analysis to evaluate the changes in subchondral bone architecture. To this end, we first examined the subchondral bone status in naive knees of 12–14-week-old mutant animals, where we observed a significantly increased bone mass in *CALCA*-deficient mice and a tendency to reduced bone mass in *αCGRP*-deficient mice (Supplementary Fig. 3). Monitoring the ACLT-induced changes during the progression of ptOA, the bone volume per tissue volume (BV/TV) of the subchondral trabecular bone was remarkably reduced in ptOA knees 4 and 8 weeks after ACLT compared to contralateral healthy controls in WT mice (Fig. 3a). Likewise, decreased trabecular numbers (Tb.N) were observed 4 weeks after ACLT, while no alternation was noticed in trabecular thickness (Tb. Th) and trabecular separation (Tb.Sp) in WT ptOA knees. Comparing outcomes in mutant mice, the ACLT-induced subchondral bone loss was highest in *CALCA*-deficient mice (Fig. 3b, c). Compared to WT and *αCGRP*-deficient mice, animals lacking *CALCA* exerted a significantly more pronounced reduction in bone volume per tissue volume, trabecular numbers and thickness 4 weeks after ACLT, which was accompanied by a pronounced increase in trabecular separation. At 8 weeks post-operatively, no significant alterations were observed in the subchondral bone loss of mutant animals, with the exception of reduced trabecular numbers in *αCGRP*-deficient mice (Fig. 3c). This is potentially explained by the fact that the severe joint deformity at this late stage of ptOA may mask phenotypical changes.

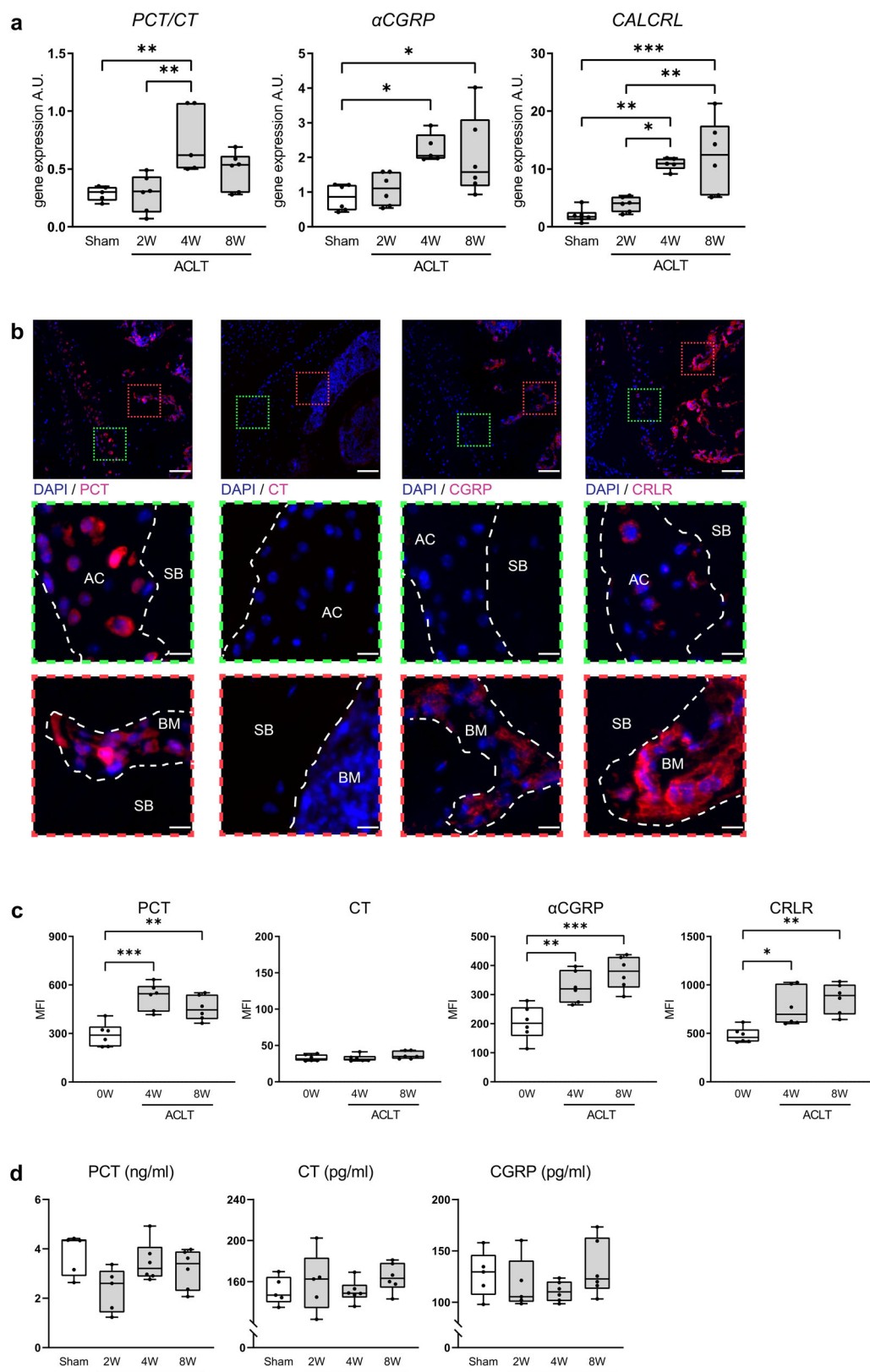

**Lack of the *PCT/CT* transcript is associated with indices of increased bone resorption in the subchondral joint compartment**

To further investigate the cellular pathomechanism underlying the observed differences in subchondral bone, histomorphometric analysis was carried out in ptOA knee sections stained with the osteoclast marker TRAP. In the subchondral trabecular bone compartment of WT mice, osteoclast surface per bone surface (Oc.S/BS) and osteoclast number per bone perimeter (Oc.N/B.Pm) were significantly increased 4 and 8 weeks after ACLT (Fig. 4a). Compared to WT mice, osteoclast numbers were remarkably increased in mice lacking *CALCA* while even decreased in *αCGRP*-deficient

**Fig. 1 | The PCT/CT and αCGRP transcripts are overexpressed during ptOA progression. a** Gene expression analysis of the *PCT/CT* and *αCGRP* transcript in addition to the CRLR (encoded by *CALCRL*) in WT knees subjected to sham or ACLT surgery and harvested at the indicated time points. *n* = 5,6 biologically independent animals per group, presented in arbitrary units (A.U.) relative to expression of GAPDH mRNA. **b** Representative overview images of immuno-fluorescent stainings specific for PCT, CT, αCGRP, and CRLR in the proximal tibia of the knee joint 4 weeks after ACLT. Site-matched images with higher magnification are presented in red and green boxes below. Subchondral bone (SB), articular cartilage (AC) and subchondral bone marrow (BM) are indicated. Scale bar = 1st row 100 μm; 2nd and 3rd rows 25 μm. **c** Quantitative evaluation of the mean fluorescence intensity (MFI) of PCT, CT, αCGRP, and CRLR in unoperated knees (0W) as well as diseased knees at 4 and 8 weeks post-operatively. *n* = 6 biologically independent animals per time point as indicated. **d** Serum concentrations of the indicated proteins in WT mice at the indicated time points after sham or ACLT surgery as measured by ELISA. *n* = 5,6 biologically independent animals per group as indicated per time point. Ordinary one-way ANOVA, box plots represent median with minimum and maximum whiskers. *$P < 0.05$, **$P < 0.01$, ***$P < 0.001$.

mice 4 weeks after ACLT (Fig. 4b, c). Similarly, highest osteoclast numbers were observed in *CALCA*[−/−] mice 8 weeks after ACLT compared to WT and *αCGRP*-deficient mice, while a trend towards decreased osteoclast parameters (not reaching statistical significance) was noticed in *αCGRP*-deficient mice in comparison to WT controls (Fig. 4c). To control for a confounding effect to due pain-induced unloading of the ptOA knee as a potential explanation for the observed alterations, we finally performed static and cellular histomorphometry of the metaphyseal compartment in the tibia of WT mice. Here, neither changes in structural bone parameters nor osteoclast surface or numbers were observed, indicating that, in line with previous reports[17,18], ACLT does not cause disuse-induced bone alterations (Supplementary Fig. 4a, b).

### Tibial osteophyte formation in ptOA is critically promoted by αCGRP signaling

The occurrence of osteophytes is one of the hallmark features of ptOA progression. To investigate the impact of the *CALCA* gene on osteophyte formation, we first evaluated the total, femoral, and tibial volume of osteophytes radiologically (Fig. 5a). Total osteophyte volume was significantly decreased in both *CALCA*- and *αCGRP*-deficient mice at 4 weeks and 8 weeks after ACLT (Fig. 5b). Although femoral osteophyte volume was not affected, the same observation was made in the volume of tibial osteophytes at both time points studied. Next, we carried out semi-quantitative histological scoring of osteophyte maturity and size (Fig. 5c). Similar to the radiological evaluation, highest scores were observed in WT mice at both time points while osteophyte formation in both *CALCA*- and *αCGRP*-deficient mice were reduced to a similar extent in the total joint and the tibial compartment (Fig. 5d).

### Reduced synovitis in *CALCA*- but not αCGRP-deficient mice

Synovitis is considered as an independent driver of ptOA onset and a promoting factor of structural deterioration. To understand the role of the *CALCA* gene on synovial inflammation, we assessed synovial hyperplasia histologically using the synovitis scoring system (Fig. 6a). *CALCA*[−/−] mice exhibited a significantly decreased synovitis score in the total joint compared to both WT and *αCGRP*-deficient mice 4 and 8 weeks after ACLT, while it remained unchanged in *αCGRP*[−/−] mice compared to WT controls (Fig. 6b). No alterations were observed in the femoral synovial inflammation in mutant animals at both time points. However, *CALCA*-deficient mice exerted a reduced tibial synovitis score 8 weeks after ACLT compared to WT mice, while no significant difference was noticed between WT and *αCGRP*-deficient animals.

### Discussion

Based on the observation that polymorphisms in the *CALCA* gene are associated with the onset and progression of osteoarthritis, this study was designed to test a functional relevance of both the *PCT/CT* and *αCGRP* transcript in a mouse model of ptOA. Our data show that the *PCT/CT* transcript is crucially involved in cartilage degeneration, abnormal subchondral bone remodeling, and synovitis, whereas the *αCGRP* transcript is a potent driver of osteophyte formation in ptOA progression (Supplementary Fig. 5).

Degenerative joint diseases, including ptOA of the knee, are chronic health conditions representing a major cause of pain, physical disability, and decreased quality of life. Despite considerable research efforts, therapeutic breakthroughs could not be achieved to date, and surgical replacement of the diseased joint remains the only definite treatment option for affected patients[6]. Based on their impact on bone and cartilage turnover in health and disease[19–21], a role of *CALCA*-encoded peptides in degenerative joint disease has long been suggested. Both genetic and pharmacological blockade of (α) CGRP signaling was reported to attenuate cartilage degeneration and pain perception in experimental ptOA[22], which however could not be confirmed in a prospective clinical trial[23]. Similarly, application of recombinant CT has been reported to protect from cartilage degeneration in experimental ptOA[24,25], while oral salmon calcitonin did not provide reproducible clinical benefits in patients with symptomatic knee osteoarthritis[26]. In addition to these conflicting results, an impact of the CT precursor, namely PCT, on ptOA has been largely neglected. A role of PCT in degenerative joint disease however is conceivable, as, similar to rheumatoid arthritis[19], inflammatory processes are central to disease progression, and as we previously showed that PCT potentiates pro-inflammatory signaling events in macrophages while inhibiting macrophage migration and fusion into osteoclasts during teriparatide therapy[14,15].

To rigorously test a potential, pathophysiologic involvement of the *PCT/CT* and the *αCGRP* transcript in ptOA, we thus employed *CALCA*- and *αCGRP*-deficient mice to ACLT and compared disease progression to WT controls. As *CALCA*[−/−] animals lack all three *CALCA*-encoded peptides (PCT, CT, and αCGRP), the comparison to *αCGRP*-deficient mice allows to delineate the function of both *PCT/CT* and *αCGRP*-transcripts. Using primers specific for both transcripts, we observed an induction of *PCT/CT* mRNA at 4 weeks and *αCGRP* mRNA at 4 and 8 weeks in ptOA knees of WT mice. These results were confirmed on protein level, where immunofluorescence showed strong signal intensities in ptOA knees stained with either a PCT- or an αCGRP-specific antibody 4 and 8 weeks after ptOA-induction. In contrast, no specific expression of the CT protein was detected on mRNA and protein level, indicating that gene transcription of the *PCT/CT* transcript primarily results in PCT protein production during ptOA. This may be explained by the fact that bone and cartilage cells, unlike thyroid C-cells, are unlikely to have the enzymatic machinery to convert PCT to CT[27]. A potential impact of PCT and αCGRP was further supported by the finding of a robust induction of CRLR on mRNA and protein level during ptOA, which mediates the biological actions of both ligands[14,28]. Together, as systemic levels of all three *CALCA*-encoded peptides remained unaltered after ACLT, these findings imply that local expression of PCT and CGRP, but not CT, might be involved in the pathophysiology of ptOA progression.

Assessing joint degeneration using OARSI histopathological grading, we found *CALCA*-deficient mice to display enhanced cartilage damage, especially 4 weeks after ACLT. Similarly, *CALCA*-deficient mice displayed a pronounced reduction in subchondral bone plate thickness, which was not the case in *αCGRP*-deficient animals. In the subchondral bone compartment, a significantly increased bone loss was observed exclusively in mice lacking *CALCA* 4 weeks after ACLT, which was associated with increased osteoclast numbers. In contrast, *αCGRP*-deficient mice did not demonstrate enhanced subchondral bone loss and even displayed reduced osteoclast parameters compared to WT with ptOA. Although we cannot rule out a direct effect of systemic CT, these findings point towards a local function of PCT to promote chondrocyte integrity and limit subchondral bone loss by inhibiting excessive osteoclastogenesis. This is in line with our previous

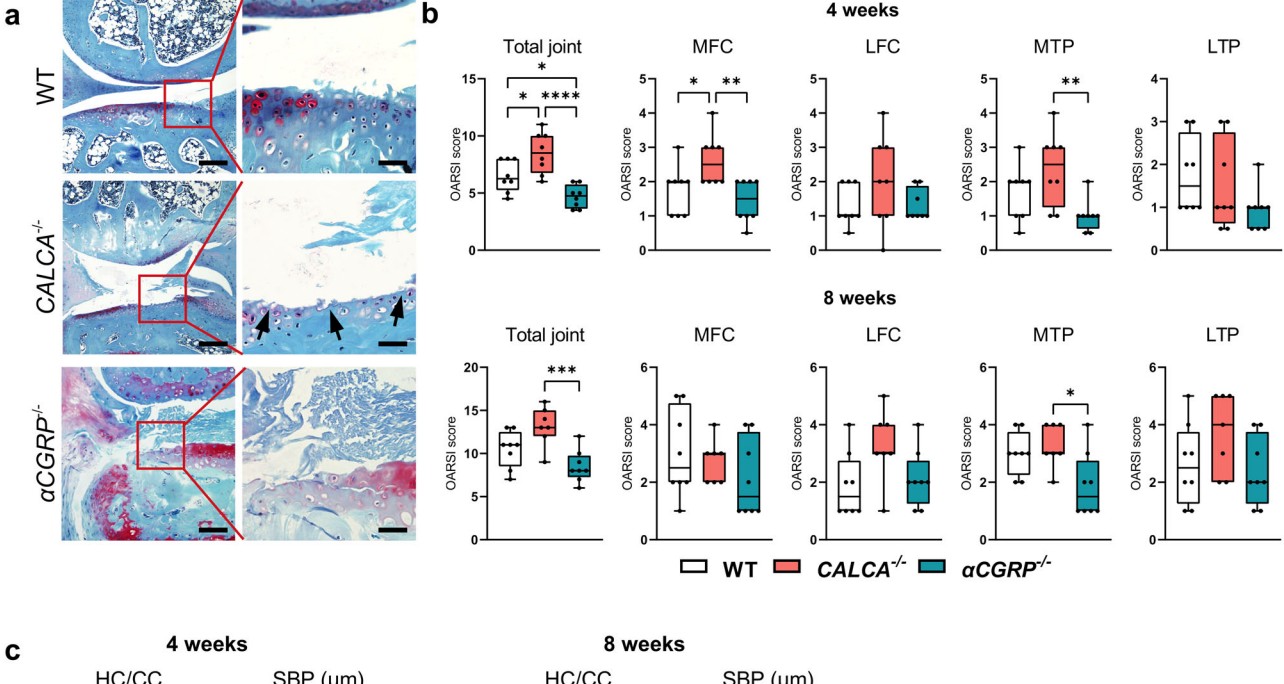

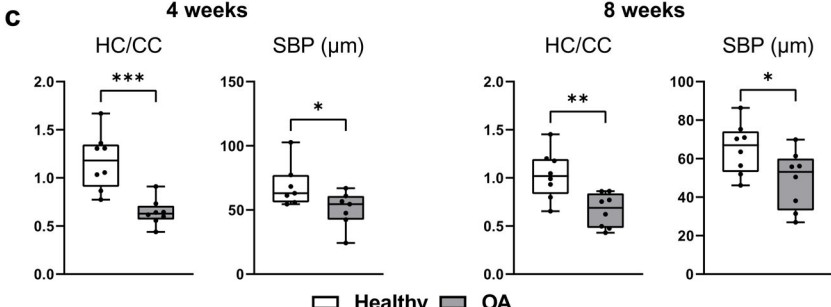

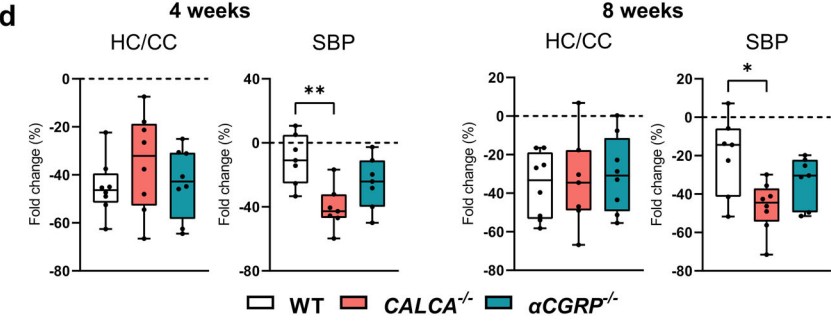

**Fig. 2 | *PCT/CT*-deficiency aggravates cartilage degeneration after ACLT.**
**a** Representative images of medial knee joints stained with BIC 4 weeks after ACLT (scale bar = left column 200 μm; right column 50 μm). The articular cartilage stains red and the bone stains blue. The arrows indicate enhanced cartilage ulceration in *CALCA*-deficient mice. **b** Osteoarthritis Research Society International (OARSI) scoring of total joints as well as of each quadrant, including the medial tibial plateau (MTP), the medial femoral condyle (MFC), the lateral tibial plateau (LTP), and the lateral femoral condyle (LFC) at the indicated time points. **c** Quantitative analysis of the subchondral bone plate thickness (SBP) and the ratio of hyaline cartilage thickness to calcified cartilage thickness (HC/CC) in healthy and diseased knees from WT mice 4 and 8 weeks after ACLT. **d** Relative alterations at the indicated timepoints (ACLT vs. contralateral healthy controls) in the SBP and HC/CC. *n* = 7,8 biologically independent animals as indicated per group and time point. For **b** and **d** ordinary one-way ANOVA was used and for **c** unpaired two-tailed students t-test was used to determine differences. Box plots represent median with minimum and maximum whiskers. *$P < 0.05$, **$P < 0.01$, ***$P < 0.001$, ****$P < 0.0001$.

observations that the primary function of systemic CT is to impair bone formation by osteoblasts, whereas local release of PCT within the bone microenvironment controls bone resorption[13,15].

In terms of formation of new bone, ptOA is characterized by the formation of osteophytes, representing a radiological hallmark of late-stage degenerative joint disease[29]. We observed a significantly decreased osteophyte volume in both *CALCA*- and *αCGRP*-deficient ptOA knees compared to WT controls. Likewise, semi-quantitative scoring of histological knee

sections showed a reduced osteophyte size and maturity in *CALCA*$^{-/-}$ and *αCGRP*$^{-/-}$ mice. As this osteophyte phenotype was observed in both mutant mouse lines, it can be concluded that this effect occurs specifically due to the lack of the *αCGRP* transcript. Osteophyte formation, initiated by mesenchymal cells originating in the periosteum, is considered to closely resemble bone regeneration in fracture healing[30]. Other authors[31] and we[32] have previously shown that αCGRP is essential for adequate bone regeneration in murine fracture models through promoting callus mineralization.

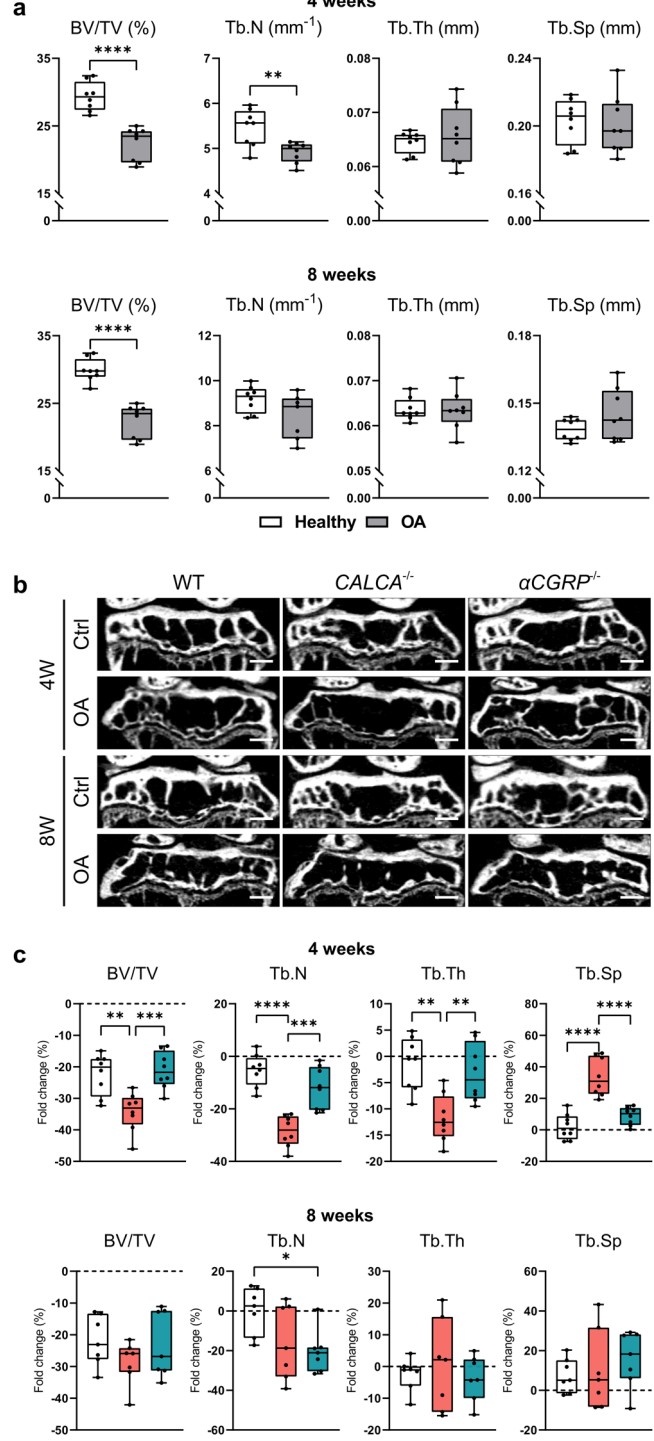

**Fig. 3 | Lack of the *PCT/CT* transcript exacerbates loss of subchondral trabecular bone. a** Micro-CT-based, quantitative analysis of tibial subchondral trabecular bone of the bone volume fraction (BV/TV), trabecular numbers (Tb.N), trabecular thickness (Tb.Th), and trabecular separation (Tb.Sp) at the indicated time points in WT mice after sham or ACLT surgery. **b** Representative micro-CT coronal views of tibial subchondral trabecular bone of diseased (OA) and contralateral control (Ctrl) knees from the indicated groups 4 and 8 weeks after ACLT (scale bar = 500 μm). **c** Relative alterations at the indicated time points (ACLT vs. contralateral healthy controls) in BV/TV, Tb.N, Tb.Th and Tb.Sp, expressed as fold differences. *n* = 7,8 biologically independent animals as indicated per group and time point. For **a** unpaired two-tailed students t-test and for **c** ordinary one-way ANOVA was used and to calculate differences. Box plots represent median with minimum and maximum whiskers. \**P* < 0.05, \*\**P* < 0.01, \*\*\**P* < 0.001, \*\*\*\**P* < 0.0001.

Although a randomized clinical phase III trial[23] showed that neutralizing CGRP did not affect joint pain in patients with osteoarthritis, pharmacological targeting of CGRP or its receptor may thus result in reduced osteophyte formation and maturation.

Assessing inflammation of the synovial membrane, another pathophysiologic feature of advanced ptOA, we observed a significantly decreased synovitis score exclusively in mice lacking *CALCA* 4 and 8 weeks after ACLT. Synovitis is present at the early stage of ptOA, persists throughout ptOA development[33], and was reported to precede the occurrence of cartilage and bone pathologies in osteoarthritis[34,35]. Therefore, synovitis may independently augment disease progression in ptOA. In this regard, we previously showed that PCT augments pro-inflammatory signaling events in activated macrophages, contributing to tissue inflammation and damage during systemic inflammation[14]. Similar mechanism of action are conceivable in the setting of ptOA, where activated synovial cells and macrophages in the inflamed synovium were shown to produce pro-inflammatory mediators and proteolytic enzymes, ultimately resulting in cartilage degradation[36]. While the reduced synovitis in *CALCA*-deficient mice is most likely explained by the lack of pro-inflammatory PCT signalling, it appears unrelated to the increase in articular cartilage degeneration in this mouse line. Thus, the *PCT/CT* transcript exerts divergent effects on the different joint compartments in ptOA, characterized by i) a protection from cartilage degeneration, ii) a reduction in subchondral bone loss, and iii) an increased degree of synovitis. It is important to emphasize not only that the major findings observed at 4 weeks are relevant to the progression of ptOA, but also that the findings at 8 weeks are more clinically relevant when translating to clinical scenarios and potential therapeutic strategies. The less pronounced phenotypes, especially in *CALCA*-deficient mice 8 weeks after ACLT, may be explained by the fact that the severe joint deformity at this late stage of ptOA potentially masks objectively measurable changes, requiring further investigations.

In this regard, the current study has several limitations. First, a major limitation is that we did not assess pain outcomes in the respective mouse lines, because these measurements are not routinely performed in our laboratory. As the ACLT model is associated with changes in pain behavior, and given the importance of nociception in ptOA and the role of αCGRP in nociception, future studies are warranted to address a corresponding influence on ptOA progression. Second, the applied experimental ptOA model is not capable of fully mimicking the complex disease progression in patients[37]. Also, different results may be obtained in alternative models such as destabilization of the medial meniscus, which usually yields a less progressive form of ptOA[38]. Third, this study was conducted with female mice, and it remains unknown whether similar effects are observable in male mice. Fourth, it is also crucial to study the role of *CALCA*-encoded peptides in primary, non-traumatic osteoarthritis, as it represents the most common cause of degenerative joint disease and exhibits important pathophysiological and biomechanical differences from ptOA. Finally, using *CALCA*⁻/⁻ and *αCGRP*⁻/⁻ mice, we cannot rule out an effect of circulating CT. A number of studies[39,40] suggested a beneficial effect of recombinant CT on chondrocyte metabolism and OA progression, while two phase III trials[41,42] reported that salmon CT did not provide reproducible clinical benefits in patients with symptomatic osteoarthritis. Thus, despite the specific expression pattern of the *PCT/CT* transcript, suggesting a direct impact of PCT, but not CT, on ptOA, further studies are warranted to delineate CT-dependent signaling cascades in ptOA.

In conclusion, this study tested the previously reported association of degenerative joint disease with polymorphisms in *CALCA*, a well-defined gene and subject to tissue-specific, alternative splicing. Our data show that the *PCT/CT* transcript protects from cartilage degeneration and subchondral bone loss in ptOA, yet aggravates inflammation of the synovial membrane. Further, this study identifies *αCGRP* as a potent driver of osteophyte formation in ptOA, at least in mice. Further studies are warranted to test whether targeting *CALCA*-encoded peptides may be beneficial to alter the course of ptOA.

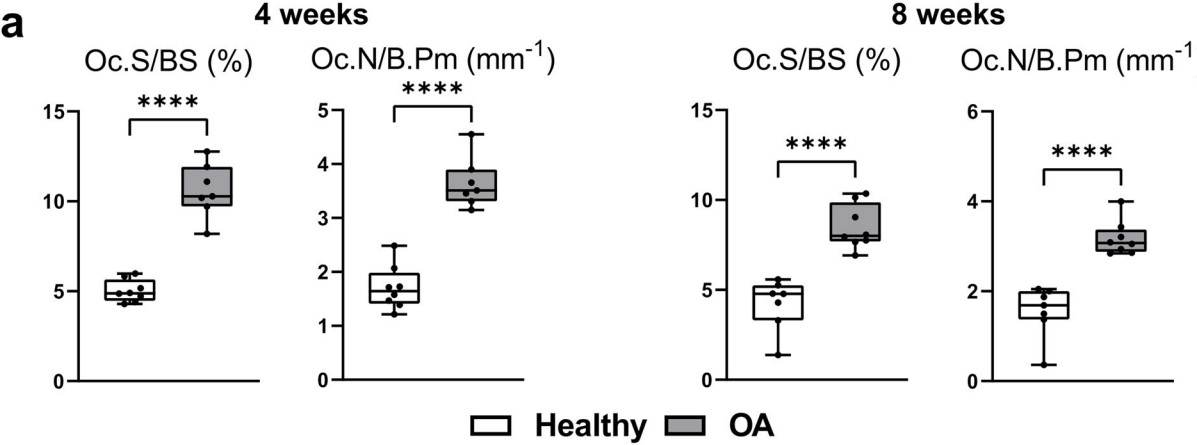

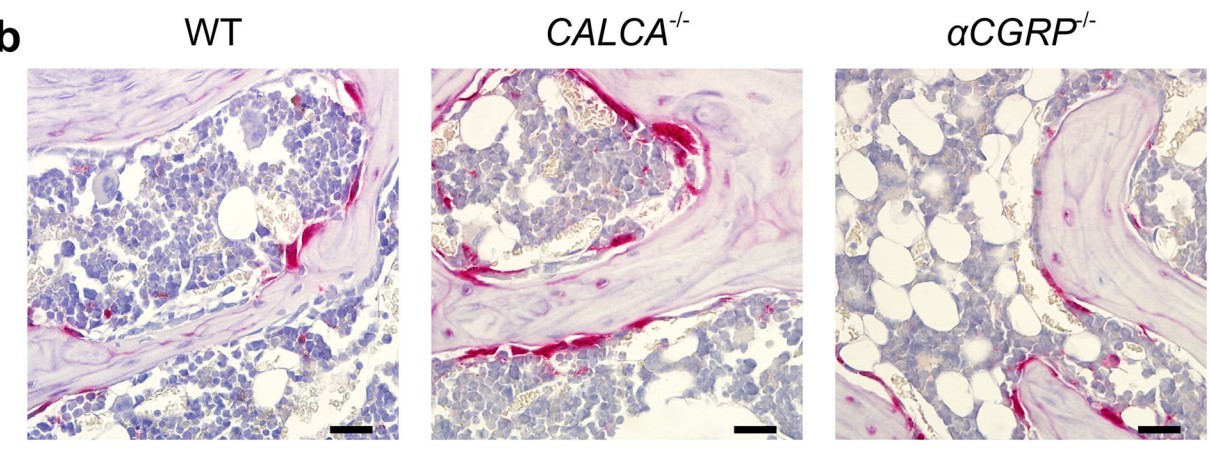

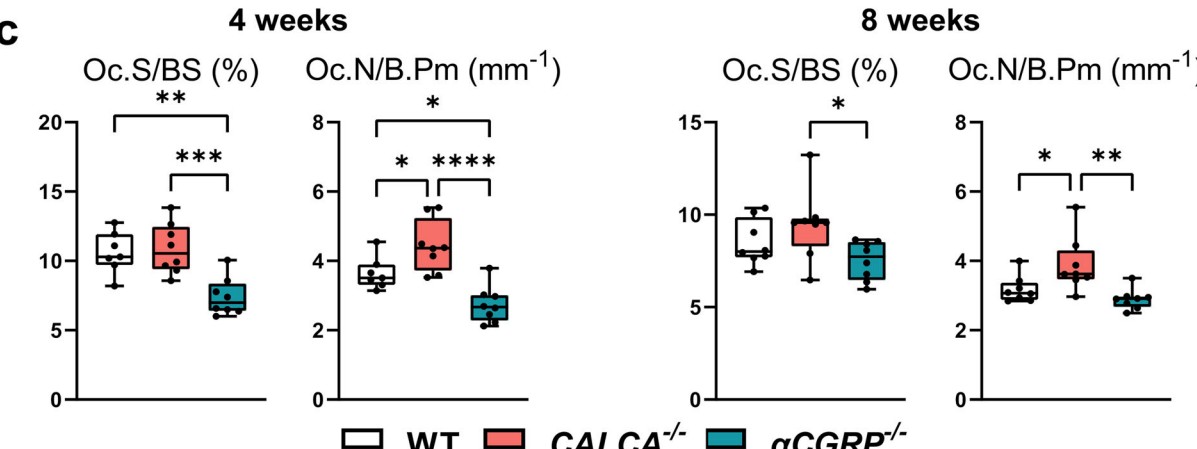

**Fig. 4 | Mice lacking the *PCT/CT* transcript display indices of increased bone resorption in subchondral bone after ACLT. a** Histomorphometric analysis of osteoclast surface per bone surface (Oc.S/BS) and numbers of osteoclasts per bone perimeter (Oc.N/B.Pm) at the indicated time points in TRAP-stained ptOA sections. **b** Representative TRAP-stained images of tibial subchondral bone of ptOA knees 4 weeks after ACLT (scale bar = 50 μm). **c** Quantification of Oc.S/BS and Oc.N/B.Pm in mutant and WT mice at the indicated timepoints. *n* = 7,8 biologically independent animals as indicated per group and time point. For **a** unpaired two-tailed students t-test and for **c** ordinary one-way ANOVA was used to determine differences. Box plots represent median with minimum and maximum whiskers. *$P < 0.05$, **$P < 0.01$, ***$P < 0.001$, ****$P < 0.0001$.

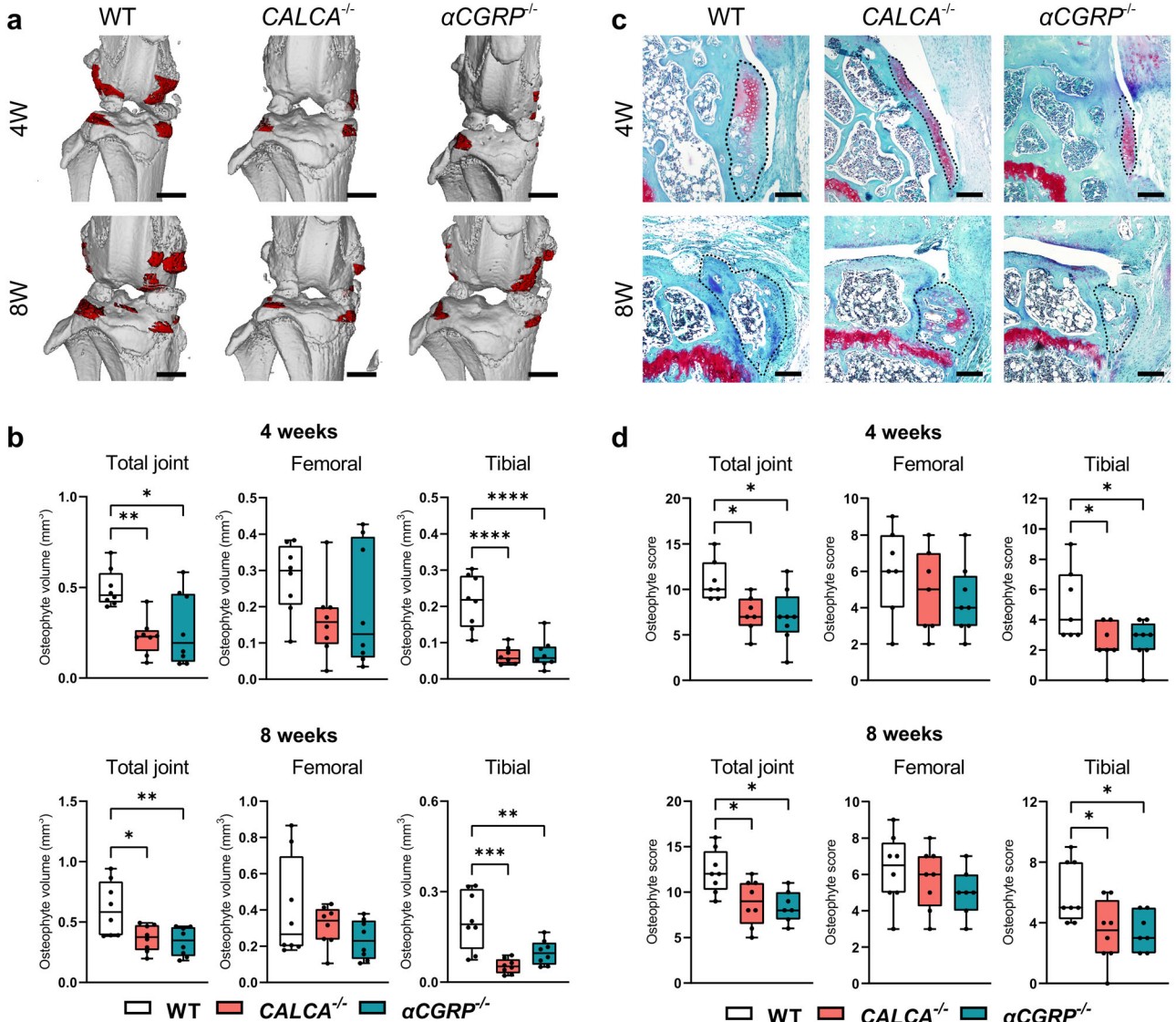

**Fig. 5 | Inactivation of the *αCGRP* transcript results in reduced osteophyte formation. a** Representative 3-dimensional micro-CT reconstruction of the osteophytes (red) in ptOA knees from the indicated groups 4 and 8 weeks after surgery (scale bar = 1 mm). **b** Quantitative evaluation of osteophyte volume in the total joint as well as femoral and tibial compartments at the indicated time points.
**c** Representative histological images of femoral osteophytes from indicated groups stained with BIC at indicated time points (scale bar = 200 μm). **d** Semi-quantitative grading of the maturity and size of osteophytes in the total joint, femoral and tibial compartments at the indicated time points. *n* = 7,8 biologically independent animals as indicated per group and time point. Ordinary one-way ANOVA was used to determine statistical differences. Box plots represent median with minimum and maximum whiskers. *$P < 0.05$, **$P < 0.01$, ***$P < 0.001$, ****$P < 0.0001$.

## Materials and methods
### Animals and ptOA induction
*CALCA⁻/⁻*, *αCGRP⁻/⁻* and WT controls, all with a C57BL/6J background, were genotyped as previously described[15,43,44]. We have complied with all relevant ethical regulations for animal use. All in vivo mouse experiments were performed in accordance with the current recommendations of the "Report of the American Veterinary Medicine Association Panel on Euthanasia" and with approval from the "Behörde für Justiz und Verbraucherschutz der Freien und Hansestadt Hamburg" (N003/2020, N101/2021). Mice were housed in a specific pathogen-free animal facility and maintained on standard conditions at a 12 h light/dark cycle and fed ad libitum.

Female mice at the age of 12-14 weeks were subjected to ACLT to induce ptOA. Briefly, after anesthesia induced by isoflurane inhalation, the right knee joint was exposed via a medial parapatellar incision. The patella was dislocated to expose the ACL. Following the transection of the ACL with a micro-surgical scalpel, the destabilization was confirmed by a positive anterior drawer test. The wound was closed with layered

suturing. Pre-operatively, 150 mg kg⁻¹ clindamycin and 0.1 mg kg⁻¹ buprenorphine were administered for the prevention of infection and provision of analgesia, respectively. After surgery, mice were placed in a recovery rack overnight and received drinking water with 1 mg ml⁻¹ metamizole for 3 days. At 4 and 8 weeks after surgery, mice were sacrificed by cardiac exsanguination under anesthesia. Both knee joints were harvested for further analysis. For some experiments, naive healthy knees of female mice (12-14 weeks of age) of each genotype were used as indicated (hereafter referred to as 0 weeks).

### Micro-computed tomography (micro-CT)
Knee samples were fixed in 3.5% formafix for 24 h. Afterwards, micro-CT scanning was carried out using Scanco vivaCT 80 (SCANCO Medical, Brüttisellen, Switzerland) with a voxel size of 15.6 μm at 70 kVp, 113 μA and 400 ms integration time. μCT Ray V4.0-4 (Scanco Medical AG, Brüttisellen, Switzerland) and imageJ were used to generate the representative images. For subchondral trabecular bone and osteophyte evaluation, the volume-of-

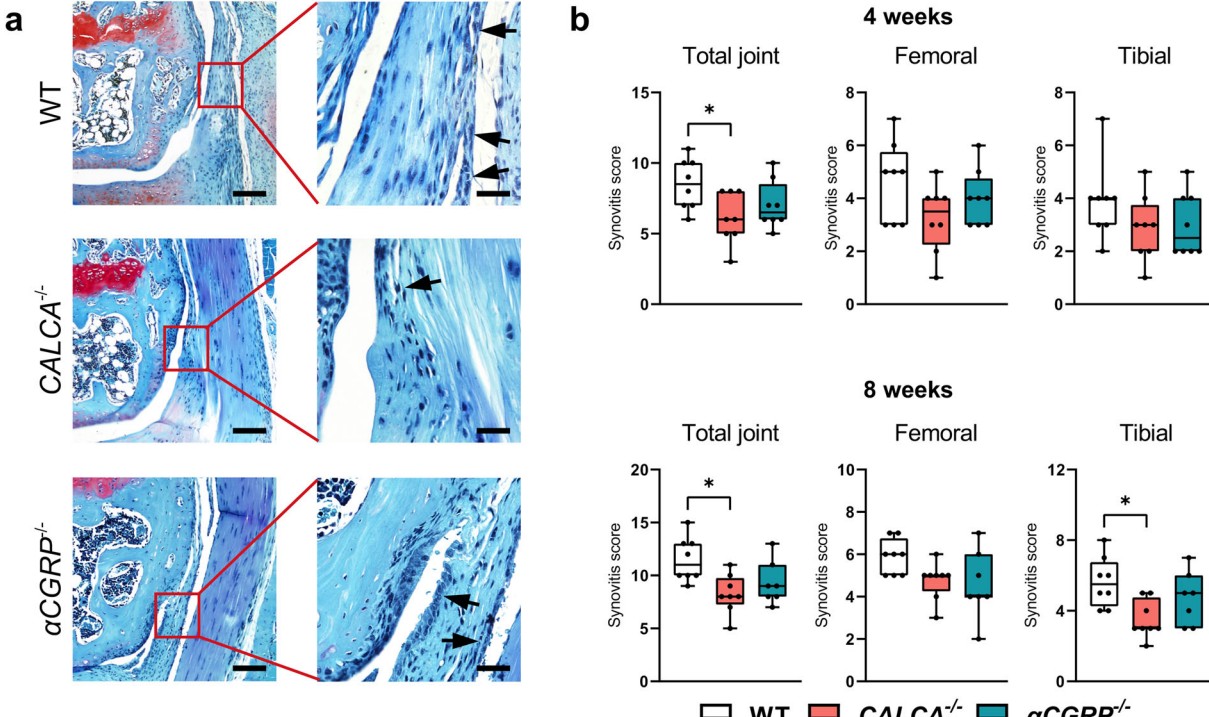

**Fig. 6 | The absence of the *PCT/CT* transcript of the *CALCA* gene alleviates ACLT-induced synovitis. a** Representative images of femoral synovitis evidenced by synovial hypertrophy and hyperplasia (black arrows) in indicated groups stained with BIC (scale bar = left column 200 μm; right column 50 μm). **b** Synovitis scoring of the total joint, femoral and tibia compartments at the indicated time points. *n* = 7,8 biologically independent animals as indicated per group and time point. Ordinary one-way ANOVA was used to determine statistical differences. Box plots represent median with minimum and maximum whiskers. **P* < 0.05.

interest (VOI) was defined by manual contouring and calculated using the μCT Evaluation Program V6.6 (Scanco Medical AG, Brüttisellen, Switzerland). Data were reported according to the guidelines for tissue imaging by the American Society of Bone and Mineral Research[45].

### Histology

Following micro-CT scanning, knee joints were decalcified in 0.5M EDTA solution (pH 7.4) for 1 week at 4 °C. Subsequently, the tissue was dehydrated, embedded in paraffin, and sectioned in a coronal plane by 4 μm thickness. Hematoxylin and eosin (H&E), Bone-Inflammation-Cartilage (BIC) stain[46], as well as tartrate-resistant acid phosphatase (TRAP) staining, were carried out. The images were acquired using a BX50 microscope connected with a DP72 camera (Olympus Optical Co., LTD, Tokyo, Japan). An area of 400 μm centered on the medial tibial plateau were defined on sections with H&E staining and the thickness of hyaline cartilage (HC), calcified cartilage (CC) and subchondral bone plate (SBP) were measured[47]. Osteoclasts were identified using TRAP staining (TRAP-positive cells with > 3 nuclei and adherent to the bone surface) and osteoclast parameters in subchondral trabecular bone were measured using the OsteoMeasure system (Osteometrics Inc., Decatur, USA). Semi-quantitative histopathological analysis using Osteoarthritis Research Society International (OARSI) score (range 0-6)[48] was carried out to evaluate cartilage degeneration. Briefly, 10 serial sections over the whole knee joint were stained with BIC, and a maximum score was assigned for each quadrant of the knee joint including the medial tibial plateau (MTP), the medial femoral condyle (MFC), the lateral tibial plateau (LTP), and the lateral femoral condyle (LFC) (Supplementary Fig. 6). The summed maximum scores of the whole joint as well as the maximum scores of the MTP, MFC, LTP and LFC were plotted. The synovitis score[49] was used to evaluate the enlargement of the synovial lining cell layer and cell density of synovial stroma at the synovial insertion of the lateral femur, medial femur, lateral tibia, and medial tibia. The maturity and size of the osteophytes on the femora and tibiae were scored to evaluate osteophyte formation[50].

### Gene expression analysis

An independent set of WT animals (C57Bl/6) was employed for gene expression analysis. The mice were subjected to ACLT and knee joints were harvested at 2,4 and 8 weeks post-operatively. After sacrifice, the joints were stripped of the skin, subcutaneous tissue and muscle. Thereafter, a 3mm long piece of the whole joint tissue including synovial membrane, articular cartilage, and subchondral bone between the distal femoral and proximal tibial growth plates was dissected. For controls, sham-operated WT mice were employed and the knee joints were collected 2 weeks after the induction. All samples were processed using a standardized RNA isolation protocol. RNA was further purified using a NucleoSpin RNA kit (Macherey-Nagel, Düren, Germany) and quantified using Nanodrop 2000 (Nanodrop Technology, MA, USA). Complementary DNA (cDNA) was then synthesized using the ProtoScript First Strand cDNA Synthesis Kit (New England BioLabs). Real-Time Quantitative Reverse Transcription PCR (qRT-PCR) was performed using TaqMan Assay-on-Demand primer sets (Applied Biosystems by Thermo Fisher Scientific, Waltham, Massachusetts, USA). Glyceraldhyde-3-phosphate dehydrogenase (*Gapdh*) was used as housekeeping gene, and the expression of target genes was presented in arbitrary units (A.U) relative to the expression of *Gapdh* mRNA.

### Immunofluorescence

To localize the cells expressing PCT, CT, αCGRP, and CRLR in diseased knee joints, independent groups of mice of each genotype (C57Bl/6J) were employed. In WT mice, the operated knees were dissected 4 and 8 weeks after ACLT and processed according to a modified Kawamoto frozen section protocol[32,51]. Naive knees of WT mice were used as healthy controls (0 weeks). To confirm successful protein inactivation in mutant mice, ptOA knees of *CALCA⁻/⁻* and *αCGRP⁻/⁻* mice were employed 4 weeks after surgery. Obtained sections were blocked in 3% BSA/5% Donkey Serum/PBS and incubated with anti-PCT (1:150, LSBio, LS-C296040), anti-CT (1: 200, Invitrogen, PA5-16464), anti-CGRP (1:300, Abcam, ab47027), and

https://doi.org/10.1038/s42003-024-05889-0 **Article**

anti-CRLR (1:200, Bioss Antibodies, bs-1860R-TR) overnight as indicated. Following subsequent washing, a secondary antibody (1:500, Life technologies, A21098) was applied and Fluromount-G with DAPI (Thermo Fisher Scientific, 00-4959-52) was used for mounting. Images were acquired using a spinning disk confocal microscope (Aurox Ltd, Oxfordshire, UK). Quantification of mean fluorescence intensity (MFI) was performed using ImageJ.

### Enzyme-Linked Immunosorbent Assay (ELISA)
Serum was collected from the same group of WT mice used for gene expression analysis. Commercial ELISA kits were applied to measure the concentration of PCT (CSB-E10371m, CUSABio, Houston, USA), CT (LS-F23047, LSBio, Seattle, USA), and CGRP (LS-F37469, CUSABio, Houston, USA) according to the manufacturers' instructions.

### Statistics and reproducibility
The sample size was calculated for the main outcome parameters derived by radiological and histological analysis, yielding a minimum group number of $n = 5$ mice. In this respect, due to the usual test approaches and laboratory conditions as well as the current lack of prior information (lack of prior probability), realistic error sizes and case numbers were assumed and, with the acceptance of type I and II of 0.05 and 0.2, respectively, group sizes were determined using non-parametric test methods allowing the evidence of effect sizes between 1.5 and 1.8 with a standard deviation of 20%. Due to occasional loss of samples during histologic processing, $n = 6$–8 per group and time point were used for the main study. A total number of 102 mice were used, including 24 WT mice for the gene expression analysis (6 mice per timepoint; 2, 4 and 8 weeks post-operatively and sham group), 18 WT and 12 mutant mice for immunofluorescence (6 mice per timepoint and colony, 0, 4 and 8 weeks post-operatively for WT mice, with additional 4-week timepoint for $CALCA^{-/-}$ and $\alpha CGRP^{-/-}$ mice) and 48 mice (3 colonies, 2 timepoints and 8 mice per group) for the main study. Groups were assigned randomly, and researchers were blinded during sample processing and analyses. Statistical analyses were performed using Graph-Pad Prism version 9.1.1 (GraphPad Software Inc., La Jolla, USA). Unpaired two-tailed students t-test or one-way ANOVA for multiple-group comparisons were used as indicated. Differences were considered statistically significant at $P < 0.05$.

### Reporting summary
Further information on research design is available in the Nature Portfolio Reporting Summary linked to this article.

### Data availability
All relevant data of this project is presented within the figures of this manuscript. Numerical source data for figures in the manuscript can be found in the Supplementary Data file.

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

## Acknowledgements
S.J. is funded by the China Scholarship Council. T.M. is participant in the BIH Charité Clinician Scientist Program funded by the Charité—Universitätsmedizin Berlin, and the Berlin Institute of Health at Charité (BIH).

## Author contributions
S.J. planned and performed experiments, analysed, and interpreted the data, and wrote the manuscript. W.X., P.K., A.D. and L.A. assisted with surgeries and tissue sampling. J.S. conducted the gene expression and data analysis. A.H., T.M. and K-H.F. provided critical discussions. C.E., J.A. and M.F. performed histological embedding, cutting and staining. A.B. designed the study and interpreted the data. J.K. designed and supervised the study, interpreted the data, and wrote the manuscript. All authors approved the final version of the manuscript.

## Funding

## Competing interests
The authors declare no competing interests.
