## [Peer Review File · Communications Biology]

Reviewers' comments:

Reviewer #1 (Remarks to the Author):

The authors present a nice paper on the role of Calcitonin in the development of osteoarthritis utilising the ACL transection model on female mice. Because clinical trials have not replicated preclinical studies with calcitonin or CGRP blockade, the aim is to understand the effect of the different polymorphisms of the CALCA gene in OA development. For this, two KO mice, developed previously, have a deletion of the whole CALCA gene or just the CGRP transcript, thus separating the PCT/CT from the CGRP. The authors conclude that PCT/CT transcript protects from cartilage degeneration and subchondral bone loss, while CGRP drives osteophytogenesis. The study is designed and written well. The data is nicely presented and the conclusions are mostly supported by the data presented and the literature.

Major concerns:

An important aspect of the evaluation of an osteoarthritis model is the OARSI score and how it is evaluated. The score should be done in multiple sections (5 to 10 sections is usual) that expand over the area of the joint where the femur and tibia are closer together. Most calculate the average, while others add the different sections (so long it is the same number of sections per sample) Looking at the graphs in figure 2, there are improvements to be made:

1) The images of the coronal sections for both the CALCA^{-/-} and the CGRP^{-/-} seem to be too early (or too late) in the joint. These areas tend to show less damage, so it could be that the damage is even higher.

2) The scoring seems to have been done only on one section, which would not be representative of the whole joint.

3) In the graphs representing the scores, there is no explanation of what they measured as the femoral or Tibial. As the authors state, there are 4 quadrants of the joint, medial tibial, lateral tibial, medial femoral and lateral femoral. Does femoral mean both lateral and medial side of the femur? In any case, it would be best to simply show the total score and the 4 different quadrants.

4) There is also a mistake in figure 2B, where the tibial at 4 weeks is the exact same graph as the tibial at 8 weeks. I doubt that the two timepoints produced the exact same data. Please correct this. The authors state that the increase in bone resorption is limited to the subchondral bone, and they compare in the WT to contralateral leg. Considering the severity of this model, in terms of joint instability, there is a chance that a trigger for the resorption is the unloading of the leg, which could then be exacerbated by the genetic modifications. Thus, it could be that the resorption is also increased in the trabecular bone of the metaphysis. There were no metaphyseal parameters measured. Therefore, we can't conclude the resorption is limited to the subchondral bone. These measurements should be done.

Minor concerns:

1) microCT resolution is low. This can not be modified now though.

2) DDCT method for qPCR calculations: This method is usually used to express a fold change in comparison to the WT (in this case). Statistics would be done on the DCT while the fold change expressed as the 2 to the -DDCT. I'm sure there is a reference for this.

3) The number of animals used:

a. There were no power calculations to justify the number of animals used. It is important to minimise the use of animals and thus plan experiments with the minimum number of animals possible (sometimes good to add one or two extra per group to count for attrition), whilst still generating a solid answer.

b. The authors state that 90 animals were used for this study and that there was an occasional loss of histology samples. By my calculations there were 5 mice per group for the knee qPCR = 20. There is no indication of how many were used for immunofluorescence. Then the main study has 8 mice per group for two different timepoints = 48. The occasional loss seems to be about 20% of the study. Please justify the numbers in the response to reviewers.

4) There should be a specification of what part of the joint was used for qPCR. The whole joint, ligaments and all? From growth plate to growth plate?

Reviewer #2 (Remarks to the Author):

In this study, authors describe the effects of CALCA and α CGRP knockout on osteoarthritis severity and features as well as subchondral bone changes in the ACLT model. The results suggest a protective role of (pro)calcitonin on subchondral bone loss and cartilage damage, while α CGRP may primarily regulate osteophyte formation.

This descriptive study may be of interest to the OA research community and corroborate data from previous experimental studies on the therapeutic efficacy of oral calcitonin supplementation or inhibition of α CGRP.

However, the initial enthusiasm for this study is diminished by a number of methodological flaws, including - but not limited to:

1) CALCA and α CGRP gene KO has not been validated. Authors refer to a schematic illustration (line 223, Supplementary Figure 1), but not actual data to support successful gene KO in bone tissue. The author's previous work (ref 34, 2008) describes clear phenotypic bone changes in naive animals, but no proof of gene expression knockout.

2) Tissue expression of procalcitonin, calcitonin, CGRP and CRLR (Figure 1B) has only been evaluated in wild type animals at week 4 post-ACLT surgery. Representative images are shown, but it is unclear how many replicates have been performed. These IHC analyses should ideally be expanded to additional time points (0 and week 8) and regions displaying remodelling in ACLT (osteophyte, degenerated articular cartilage, synovium). Finally, IHC analyses on KO animals would provide convincing evidence for gene knockout in the tissue of interest.

3) It is not appropriate to use the contralateral knee joint as healthy controls (line 238), as altered pain behaviour and gait will have an influence on the contralateral joint. A separate sham control group should be used here.

4) How do authors account for confounding effects caused by differential pain behaviour? Whole body α CGRP KO is likely to affect this in OA by mechanisms beyond local bone/osteophyte formation. Assessment of pain responses may be necessary to determine whether this is a confounding factor.

5) In their previous work (ref 34) authors described clear subchondral bone phenotypes in 12 months old mice. Did the authors verify whether these phenotypes are present in female mice aged 12-14 weeks (line 116)? A comparison of bone parameters in naive KO mice should be provided. Changes observed in the ACLT model have been reported as fold change over contralateral healthy joints, which is not appropriate (see point 3) - and it is unclear whether controls were performed for each genotype.

6) In absence of the aforementioned data, the reported results provide a set of interesting phenotypical observations - but lack the rigor to suggest targeting of CALCA-encoded peptides (line 409) in post-traumatic OA.

Jeroen Geurts Ph.D.
Lausanne University Hospital
Switzerland

Reviewer #3 (Remarks to the Author):

The objectives of the investigation was to examine the functional relevance of the PCT/CT and α CGRP transcripts in a mouse model of post-traumatic osteoarthritis (ptOA). OA is a chronic degenerative joint disease affecting cartilage, subchondral bone and soft tissues. Animal models, offer valuable insights into potential OA mechanisms in people. The subject represents a logical, topical and relevant line of inquiry, considering the established associations between polymorphisms in the CALCA gene and the onset and progression of OA in people. The genetic factors and trauma/instability, resulting from anterior cruciate ligament injury modelled are both pertinent to the study of OA mechanisms in people. It should be kept in mind that the timespan of the animal model is very short and, although the 4 week observations provide insight into the progression of the disease, the 8 week findings are the most important for translation to clinical OA in people. As there are currently there no effective drugs to prevent or arrest OA, studies such as this one, that shed light on the complex mechanisms of this disease are a valuable contribution of new knowledge.

The manuscript is clearly written and easy to understand. The introduction is outstanding and provides appropriate background information supporting the line of inquiry. The methods employed are state - of-the art, comprehensive and appropriate for the questions asked. The articular cartilage, bone (subchondral plate, osteoclasts and osteophytes) and synovium were all examined. The methods were rigorous including standard histology, histomorphometry, immunofluorescent staining of the joint tissues, gene expression, and micro-CT. Osteoclasts were also TRAP stained and counted in the subchondral bone. The expression of OA in the WT mice was first examined and subsequently the WT OA group is compared with pathology in both mutants.

An increased expression of the PCT/CT (at 4W) and α CGRP (at 4 & 8W) transcripts was detected in the ptOA WT knees. Significant effects on cartilage degeneration and subchondral bone parameters were detected at 8 weeks in the presence of the PCT/CT transcript. On the other hand, the presence of α CGRP transcript was associated with osteophyte severity.

The figures are of high quality and overall easy to understand. As there is a large volume of complex and interesting data presented it would be helpful if the authors could create an infographic to summarize the statistically significant findings at week 8 to facilitate rapid, comprehension of the pertinent by readers. It would be complimentary to the supplementary figure 1 and could be included in the main figures. It would also be informative to remind and emphasize for the readership in the discussion that the significant changes observed at 4 weeks are relevant for the progression of the disease but that the findings at 8 weeks are of greater interest when translating to clinical OA and potential therapeutic strategies.

The discussion weaves in pertinent literature and also elaborates on the main limitations of this experimental model of OA. This paper will be of interest to researchers in this field, particularly in respect to the events observed in the subchondral bone.

Specific comments

Abstract: The authors should consider employing the term micro-CT throughout the manuscript as it accurately reflects what was done and immediately lets the reader know that state of the art technology was employed to investigate bone in the mouse model.

L153 Was this a maximal score in a section? If so, please mention it. Where was the synovitis score made , as it can vary at different sites? In respect to osteophyte scores, it should be added that both femoral and tibial osteophytes were assessed.

L 157. Which tissues were employed for the gene expression studies?

L 171; L 174; L219-213. For the WT ptOA studies it seems that time points 2, 4, and 8 weeks were studied. Data from 4 and 8 weeks are subsequently presented. Was immunofluorescent staining applied to unoperated control WT mice or 8 week post ACLT WT mice? If the authors have this data, it should be presented also. This is important as the gene expression was not increased at 8 weeks. The presence of PCT in the tissues is important for the claims made in respect to its role in OA.

L 251. If possible, present site -matched representative micro-CT images of ptOA, and mutants at 4 and 8 weeks.

L329-331. It should be pointed out that this was observed at 4 weeks alone and not 8 weeks. This will then link to L 339 linking the observation to progression.

FIGURES

Figure are clear but figure legends lack information. Please provide the n values for each experiment in the figure legends.

Figure 1

A. What is A.U. in the figure?

B. Please summarize what the figures are showing in the figure legend. Symbols could be inserted in images with mention of the specific cells that were immunostained.

L 583. Please state where the sections were made, femur or tibia.

L586. n=5-6 per group as indicated per time point. Why is "as indicated" here?

Figure 2

A. Please state what the images are revealing for readers in the figure legend... meaning of pink and blue stain, articular cartilage ulceration etc.

Figure 3

Please provide pre surgery, WT, mutant representative site matched images for both 4 and 8 weeks.

Figure 5

A. can you also show micro -CT images from 8 weeks post-surgery?

Figure 6.

A. Please expand figure legend to state what you are showing in these histological images (for example for lower row hyperplasia and synovial hypertrophy are evident).

Point-to-point reply for COMMSBIO-23-1803

'The PCT/CT transcript of the CALCA gene controls cartilage degradation and subchondral bone loss, while α CGRP promotes osteophyte formation in murine post-traumatic osteoarthritis'

Reviewer #1 (Remarks to the Author):

The authors present a nice paper on the role of Calcitonin in the development of osteoarthritis utilizing the ACL transection model on female mice. Because clinical trials have not replicated preclinical studies with calcitonin or CGRP blockade, the aim is to understand the effect of the different polymorphisms of the CALCA gene in OA development. For this, two KO mice, developed previously, have a deletion of the whole CALCA gene or just the CGRP transcript, thus separating the PCT/CT from the CGRP. The authors conclude that PCT/CT transcript protects from cartilage degeneration and subchondral bone loss, while CGRP drives osteophytogenesis. The study is designed and written well. The data is nicely presented and the conclusions are mostly supported by the data presented and the literature.

Major concerns:

An important aspect of the evaluation of an osteoarthritis model is the OARSI score and how it is evaluated. The score should be done in multiple sections (5 to 10 sections is usual) that expand over the area of the joint where the femur and tibia are closer together. Most calculate the average, while others add the different sections (so long it is the same number of sections per sample) Looking at the graphs in figure 2, there are improvements to be made:

1) The images of the coronal sections for both the CALCA^{-/-} and the CGRP^{-/-} seem to be too early (or too late) in the joint. These areas tend to show less damage, so it could be that the damage is even higher.

Response: We appreciate this valuable comment from the reviewer. For OARSI scoring, we measured 10 serial coronal sections spanning the anterior, middle, and posterior portions of the joint and assigned a maximum score for each quadrant. This information is now included in the revised Methods section. For visualization and readability, the images shown represent the area with the highest degree of cartilage erosion.

Supplementary Figure 1, L152-L157.

2) The scoring seems to have been done only on one section, which would not be representative of the whole joint.

Response: We thank the reviewer for raising this critical point and apologize for the previous inaccuracy of the methodological description. In fact, we applied the 0-6 subjective scoring system to all four quadrants and via 10-step sections through the entire joint, as previously recommended by the OARSI histopathology initiative¹. As noted above, a maximum score was assigned in each section for the four quadrants of the knee joint, including the medial tibial plateau (MTP), medial femoral condyle (MFC), lateral tibial plateau (LTP), and lateral femoral condyle (LFC). OA severity is now expressed as the sum of the maximum scores of the total joint, in addition to the individual femoral and tibial quadrant scores, respectively.

Supplementary Figure 1, L152-L157.

3) In the graphs representing the scores, there is no explanation of what they measured as the femoral or tibial. As the authors state, there are 4 quadrants of the joint, medial tibial, lateral tibial, medial femoral and lateral femoral. Does femoral mean both lateral and medial side of the femur? In any case, it would be best to simply show the total score and the 4 different quadrants.

*Response: We thank the referee for the valuable comment and apologize for the lack of clarity. As suggested by the reviewer, we now show the total joint score and the score of the four different quadrants in the **new Figure 2**.*

4) There is also a mistake in figure 2B, where the tibial at 4 weeks is the exact same graph as the tibial at 8 weeks. I doubt that the two timepoints produced the exact same data. Please correct this.

*Response: We appreciate your pointing out this error and apologize for the confusion. The correction has been made to the image in the **new Figure 2**.*

The authors state that the increase in bone resorption is limited to the subchondral bone, and they compare in the WT to contralateral leg. Considering the severity of this model, in terms of joint instability, there is a chance that a trigger for the resorption is the unloading of the leg, which could then be exacerbated by the genetic modifications. Thus, it could be that the resorption is also increased in the trabecular bone of the metaphysis. There were no metaphyseal parameters measured. Therefore, we can't conclude the resorption is limited to the subchondral bone. These measurements should be done.

*Response: We appreciate the critical comment. As suggested by the reviewer, we have now performed static and cellular histomorphometric analysis of the metaphyseal trabecular bone in the tibia of WT mice at 8 weeks post-operatively to rule out the effect of altered weight-bearing pattern of the injured knee. Here we did not find any significant changes in trabecular structural bone parameters or indices of bone resorption (**new Supplementary Figure 5**). As the most likely explanation, Möller et al. reported that ACLT-operated mice showed an altered weight-bearing pattern of the hindlimb compared to sham-operated controls only in the first week after surgery, but not from 2 to 20 weeks post-operatively². Furthermore, the ACLT rodent model showed no differences in gait analysis and pain-related behavior between the two hind limbs in a further study³. Thus, the above observations strongly suggest that metaphyseal bone resorption is not affected by ACLT, indicating that the increased subchondral bone loss is not due to joint instability or unloading of the operated limb. These considerations have now been included in the new Discussion section of the revised manuscript. **L310-L315***

Minor concerns:

1) microCT resolution is low. This cannot be modified now though.

Response: We thank the reviewer for this comment and are aware that the resolution of our vivaCT 80 is not completely ideal, as it is designed to allow time-efficient in vivo measurements in living mice. However, all corresponding microCT measurements were performed using a voxel size of 15.6µm, which provides sufficient imaging efficiency with little compromise in resolution to reliably assess subchondral bone architecture and osteophyte volume. Therefore, although not providing the most perfect bone images, the applied microCT scanning provides reliable structural outcome parameters that fully support the conclusions presented in the manuscript.

2) DDCT method for qPCR calculations: This method is usually used to express a fold

change in comparison to the WT (in this case). Statistics would be done on the DCt while the fold change expressed as the 2 to the -DDCt. I'm sure there is a reference for this.

*Response: We appreciate the valuable comment and apologize for the misdescription. The present graphs were plotted using the relative expression of the target genes to Gapdh in arbitrary units (AU), with which the statistics were performed. dCt represents a log AU metric, and 2^{-dCt} is a linear AU metric (assuming 100% efficiency). We chose to use the former because qPCR data is almost always lognormally distributed. The manuscript has been modified accordingly for better understandability. **L177-L179***

3) The number of animals used:

a. There were no power calculations to justify the number of animals used. It is important to minimize the use of animals and thus plan experiments with the minimum number of animals possible (sometimes good to add one or two extra per group to count for attrition), whilst still generating a solid answer.

*Response: Thank you for the critical comment. As stated in the Statistics section, the sample size was calculated for the main outcome parameters derived by radiological and histological analysis, yielding a minimal group number of n=5 mice. The basic assumptions for these calculations are now described in more detail in the modified Statistics section. Due to occasional loss of samples during histologic processing, n=6-8 per group and time point were used for the study. **L203-L208***

b. The authors state that 90 animals were used for this study and that there was an occasional loss of histology samples. By my calculations there were 5 mice per group for the knee qPCR = 20. There is no indication of how many were used for immunofluorescence. Then the main study has 8 mice per group for two different timepoints = 48. The occasional loss seems to be about 20% of the study. Please justify the numbers in the response to reviewers.

*Response: We apologize for the confusion. In the revised study, we have now employed in total 102 mice, including 24 WT mice for the gene expression analysis (6 mice per timepoint, 0, 2, 4 and 8 weeks post-operatively), 18 WT and 12 mutant mice for immunofluorescence (6 mice per timepoint and colony, 0, 4 and 8 weeks post-operatively for WT mice, with additional 4-week timepoint for Calca^{-/-} and α CGRP^{-/-} mice) and 48 mice (3 colonies, 2 timepoints and 8 mice per group) for the main study. This information has been added to the revised manuscript. **L125-L127, L209-L214***

4) There should be a specification of what part of the joint was used for qPCR. The whole joint, ligaments and all? From growth plate to growth plate?

*Response: We apologize for the lack of details on how the tissue samples were obtained. For gene expression analysis, the joint was stripped of the skin, subcutaneous tissue and muscle. Thereafter, a 3mm long piece of the whole joint tissue including synovial membrane, articular cartilage, and subchondral bone between the distal femoral and proximal tibial growth plates was dissected. This information has been added to the revised manuscript. **L165-L169***

Reviewer #2 (Remarks to the Author):

In this study, authors describe the effects of CALCA and α CGRP knockout on osteoarthritis severity and features as well as subchondral bone changes in the ACLT model. The results suggest a protective role of (pro)calcitonin on subchondral bone

loss and cartilage damage, while α CGRP may primarily regulate osteophyte formation. This descriptive study may be of interest to the OA research community and corroborate data from previous experimental studies on the therapeutic efficacy of oral calcitonin supplementation or inhibition of α CGRP.

However, the initial enthusiasm for this study is diminished by a number of methodological flaws, including - but not limited to:

1) CALCA and α CGRP gene KO have not been validated. Authors refer to a schematic illustration (line 223, Supplementary Figure 1), but not actual data to support successful gene KO in bone tissue. The author's previous work (ref 34, 2008) describes clear phenotypic bone changes in naive animals, but no proof of gene expression knockout.

*Response: We appreciate the reviewer's comment and apologize for the confusion. Genotyping of CALCA- and α CGRP-deficient mice was performed as previously described^{4,5}. As suggested by the reviewer, and to further demonstrate successful ablation of the targets, we now performed immunofluorescent microscopy on knee sections from the mutant animals 4 weeks after ACLT, a time point when both PCT and CGRP are prominently induced in WT mice. We found no signal from any of the three CALCA-encoded peptides in CALCA^{-/-} knees. In α CGRP-deficient mice, expression of α CGRP was absent, while immunofluorescence still showed strong PCT signals in subchondral bone and articular cartilage (**new Supplementary Figure 3C**). Moreover, we now included additional references regarding the nature and genetic modifications of the respective mouse lines in the revised manuscript. **L110, L186-L187, L250-256, L535-L41***

2) Tissue expression of procalcitonin, calcitonin, CGRP and CRLR (Figure 1B) has only been evaluated in wild type animals at week 4 post-ACLT surgery. Representative images are shown, but it is unclear how many replicates have been performed. These IHC analyses should ideally be expanded to additional time points (0 and week 8) and regions displaying remodeling in ACLT (osteophyte, degenerated articular cartilage, synovium). Finally, IHC analyses on KO animals would provide convincing evidence for gene knockout in the tissue of interest.

*Response: Thank you for this important remark. We initially decided to present our immunofluorescence findings 4 weeks after ACLT-induction since we observed the most pronounced pathophysiological changes at this time point post-operatively. As pointed out by the reviewer and to further investigate the expression of CALCA-encoded peptides during the course of ptOA progression, we now carried out immunofluorescent stainings on unoperated knees (=0 weeks) and diseased joints at 8 weeks after ACLT induction. In the unoperated joints, no signal for the PCT and CT protein was observed in any region of interest, while a mild expression of α CGRP in the subchondral bone and a moderate immunofluorescence of CRLR in the articular cartilage and subchondral bone was noticed. On the contrary, a remarkably induced expression of PCT, α CGRP and CRLR was observed in diseased knees 8 weeks after the operation (**new Figure 1C, new Supplementary Figure 2A,B**). Similar to our findings at 4 weeks after ACLT, the expression of CT protein was not detected in healthy or the diseased joints at 8 weeks post-operatively. As stated in our response above, the successful gene ablation has now been confirmed in the knees of respective mutant lines 4 weeks post-operatively. For all immunofluorescent studies, n=6 replicates were performed for each time point and genotype. **L182-L187, L194, L210-L215, L228-L239, L654-L656***

3) It is not appropriate to use the contralateral knee joint as healthy controls (line 238), as altered pain behavior and gait will have an influence on the contralateral joint. A

separate sham control group should be used here.

Response: We appreciate the reviewer's critical comment. There is no doubt that the contralateral unoperated knee is affected to some extent by ACLT, making it not the most perfect control for ptOA progression. However, based on the 3R principles, we are committed to keeping the number of animals used to a minimum. In this regard, it has been reported that the ACLT model did not show clear differences in gait parameters and pain-related behaviors^{2,3}. In line with this, another study found that the OARSI scores of knees contralateral to the operated side were not significantly different from those of sham-operated knees⁶. In light of these reports, our request for approval of additional sham controls was denied by the local animal welfare authority, also because a number of previous well-performed studies have used the contralateral knee as a control without affecting their respective conclusions⁷⁻¹¹. Taking these points into consideration, and although we cannot rule out a minor effect of systematic changes such as circulating cytokines or pain transmitters, we are strongly convinced that the contralateral joint can be used as an adequate control for the ptOA knee in this study.

4) How do authors account for confounding effects caused by differential pain behaviour? Whole body α CGRP KO is likely to affect this in OA by mechanisms beyond local bone/osteophyte formation. Assessment of pain responses may be necessary to determine whether this is a confounding factor.

Response: As discussed above, and to the best of our knowledge, pain-related behaviors have not been shown to be profoundly affected by ACLT in mice. Thus, pain responses in the present model are unlikely to confound the results and would be expected to be of minor impact on the progression of ptOA despite the presence or absence of α CGRP. This is best illustrated by the fact that CALCA- and α CGRP-deficient mice have distinctly different ptOA phenotypes, despite the fact that both mouse lines lack the nociceptive neuropeptide α CGRP.

5) In their previous work (ref 34) authors described clear subchondral bone phenotypes in 12 months old mice. Did the authors verify whether these phenotypes are present in female mice aged 12-14 weeks (line 116)? A comparison of bone parameters in naive KO mice should be provided. Changes observed in the ACLT model have been reported as fold change over contralateral healthy joints, which is not appropriate (see point 3) - and it is unclear whether controls were performed for each genotype.

*Response: We appreciate the reviewer's comment and apologize for the lack of clarity. The skeletal phenotypes of CALCA^{-/-} and α CGRP^{-/-} mice have been described in detail in previous studies, although there the focus was not on subchondral but rather on trabecular bone in the spine and tibia. Female CALCA^{-/-} mice exhibited increased bone mass at the age of 1 month and 3 months⁴, whereas deficiency of α CGRP affected the overall bone architecture at a later age due to decreased bone formation⁵. As suggested by the reviewer, we have now assessed the structural bone parameters in naive KO mice. In line with previous reports, CALCA^{-/-} mice showed an increased BV/TV in the subchondral compartment at the age of 3 weeks, whereas a tendency towards a reduced bone mass was observed in mice lacking α CGRP. This information has now been added to the revised manuscript in the **new Supplementary Figure 4**, which also emphasizes the importance of presenting the respective changes as fold changes rather than absolute numbers to comprehensively describe the genotype-dependent differences in disease progression. **L280-L283***

6) In absence of the aforementioned data, the reported results provide a set of

interesting phenotypical observations - but lack the rigor to suggest targeting of CALCA-encoded peptides (line 409) in post-traumatic OA.

Response: While we respectfully disagree with some of the general concerns raised, we greatly appreciate Dr. Jeroen Geurts' thoughtful and constructive comments and would like to express our gratitude for the time and expertise he has devoted to evaluating our manuscript.

Reviewer #3 (Remarks to the Author):

The objectives of the investigation were to examine the functional relevance of the PCT/CT and α CGRP transcripts in a mouse model of post-traumatic osteoarthritis (ptOA). OA is a chronic degenerative joint disease affecting cartilage, subchondral bone and soft tissues. Animal models, offer valuable insights into potential OA mechanisms in people. The subject represents a logical, topical and relevant line of inquiry, considering the established associations between polymorphisms in the CALCA gene and the onset and progression of OA in people. The genetic factors and trauma/instability, resulting from anterior cruciate ligament injury modelled are both pertinent to the study of OA mechanisms in people. It should be kept in mind that the timespan of the animal model is very short and, although the 4-week observations provide insight into the progression of the disease, the 8-week findings are the most important for translation to clinical OA in people. As there are currently there no effective drugs to prevent or arrest OA, studies such as this one, that shed light on the complex mechanisms of this disease are a valuable contribution of new knowledge.

The manuscript is clearly written and easy to understand. The introduction is outstanding and provides appropriate background information supporting the line of inquiry. The methods employed are state -of-the art, comprehensive and appropriate for the questions asked. The articular cartilage, bone (subchondral plate, osteoclasts and osteophytes) and synovium were all examined. The methods were rigorous including standard histology, histomorphometry, immunofluorescent staining of the joint tissues, gene expression, and micro-CT. Osteoclasts were also TRAP stained and counted in the subchondral bone. The expression of OA in the WT mice was first examined and subsequently the WT OA group is compared with pathology in both mutants.

An increased expression of the PCT/CT (at 4W) and α CGRP (at 4 & 8W) transcripts was detected in the ptOA WT knees. Significant effects on cartilage degeneration and subchondral bone parameters were detected at 8 weeks in the presence of the PCT/CT transcript. On the other hand, the presence of α CGRP transcript was associated with osteophyte severity.

The figures are of high quality and overall easy to understand. As there is a large volume of complex and interesting data presented it would be helpful if the authors could create an infographic to summarize the statistically significant findings at week 8 to facilitate rapid, comprehension of the pertinent by readers. It would be complimentary to the supplementary figure 1 and could be included in the main figures. It would also be informative to remind and emphasize for the readership in the discussion that the significant changes observed at 4 weeks are relevant for the progression of the disease but that the findings at 8 weeks are of greater interest when translating to clinical OA and potential therapeutic strategies.

Response: We thank the reviewer for these positive remarks. As suggested, we have now included a graphical summary which highlights the main findings of this study (new Supplemental Figure 6). In addition, we have included into the discussion that the changes observed at 4 weeks are important for the progression of ptOA but that

the findings at 8 weeks are of greater interest when translating to clinical OA and potential therapeutic strategies. L429-L435

The discussion weaves in pertinent literature and also elaborates on the main limitations of this experimental model of OA. This paper will be of interest to researchers in this field, particularly in respect to the events observed in the subchondral bone.

Specific comments Specific comments

Abstract: The authors should consider employing the term micro-CT throughout the manuscript as it accurately reflects what was done and immediately lets the reader know that state of the art technology was employed to investigate bone in the mouse model.

Response: We appreciate the reviewer's suggestion. The text has been revised accordingly to improve the clarity. L34, L129, L130, L140, L678, L681, L702

L153 Was this a maximal score in a section? If so, please mention it. Where was the synovitis score made, as it can vary at different sites? In respect to osteophyte scores, it should be added that both femoral and tibial osteophytes were assessed.

*Response: We thank the reviewer for this remark. For cartilage degeneration, we applied the 0-6 subjective scoring system to all four quadrants and through 10-step sections through the entire joint, as previously recommended by the OARSI histopathology initiative¹ (new **Supplementary Figure 1**). As stated in our response to reviewer#1, a maximum score was assigned in each section for the four quadrants of the knee joint, including the medial tibial plateau (MTP), medial femoral condyle (MFC), lateral tibial plateau (LTP), and lateral femoral condyle (LFC). OA severity is expressed as the sum of the maximum joint, femoral, and tibial quadrant scores, respectively, and displayed in the **new Figure 2B**. Similarly, synovitis scoring was carried out at the synovial insertion of the lateral femur, medial femur, lateral tibia, and medial tibia and summed scores at indicated sites were plotted. The respective amendments have been made to the revised manuscript. L152-L157.*

L 157. Which tissues were employed for the gene expression studies?

Response: We apologize for the lack of details on how the tissue samples were obtained. For gene expression analysis, the joint was stripped of the skin, subcutaneous tissue and muscle. Thereafter, a 3mm long piece of the whole joint tissue including synovial membrane, articular cartilage, and subchondral bone between the distal femoral and proximal tibial growth plates was dissected. This information has now been added to the revised manuscript. L165-L169

L 171; L 174; L219-213. For the WT ptOA studies it seems that time points 2, 4, and 8 weeks were studied. Data from 4 and 8 weeks are subsequently presented. Was immunofluorescent staining applied to unoperated control WT mice or 8 week post ACLT WT mice? If the authors have this data, it should be presented also. This is important as the gene expression was not increased at 8 weeks. The presence of PCT in the tissues is important for the claims made in respect to its role in OA.

Response: We appreciate the reviewer's valuable comment. We initially decided to present immunofluorescence 4 weeks after the induction since we observed the most pronounced pathophysiological changes at 4 weeks post-operatively. To further investigate the expression of CALCA- encoded peptides during the course of ptOA progression, we now carried out additional immunofluorescent microscopy on

unoperated control knees (=0 weeks) and diseased joints at 8 weeks after the induction. In the unoperated joints, we did not detect a positive signal for either the PCT or the CT protein. In contrast, a mild expression of α CGRP in the subchondral bone and a moderate immunofluorescence of CRLR in the articular cartilage and subchondral bone was noticed. At 8 weeks post-operatively, a remarkably induced expression of PCT, α CGRP and CRLR was observed in ACLT knees. In the case of CT however, the expression of CT protein was undetectable in the diseased joints at 8 weeks post-operatively, similar to our observations at the 4-week time point. These findings have now been included into the revised version of the manuscript in the **new Figure 1C** and the **new Supplemental Figure 2. L182-L187,L194, L210-L215, L228-L239, L654-L656**

L 251. If possible, present site -matched representative micro-CT images of ptOA, and mutants at 4 and 8 weeks.

*Response: We thank the reviewer for this suggestion and have now included site-matched images in the **new Figure 3 and 5.***

L329-331. It should be pointed out that this was observed at 4 weeks alone and not 8 weeks. This will then link to L 339 linking the observation to progression.

*Response: We agree with the reviewer and have included the respective amendment, also in respect to the new immunofluorescent measurements on ptOA knees 8 weeks after ACLT. **L370-L372***

FIGURES

Figure are clear but figure legends lack information. Please provide the n values for each experiment in the figure legends.

*Response: Modified according to the reviewer's remark. **L649***

Figure 1

A. What is A.U. in the figure?

*Response: We apologize for the confusion. The expression of target genes was presented in arbitrary units (A.U.) relative to expression of GAPDH mRNA. The respective modifications to the figure legends have been made. **L649***

B. Please summarize what the figures are showing in the figure legend. Symbols could be inserted in images with mention of the specific cells that were immunostained.

*Response: We thank for the valuable suggestion. Detailed information of the figure has been added to the revised manuscript. The employed staining protocol does not allow to clearly identify the nature of the stained cells, as we focused on whether or not the respective CALCA-encoded peptides are expressed in the diseased joint or not. However, we have now indicated the particular compartment in which the signal was observed including subchondral bone (SB), articular cartilage (AC) and subchondral bone marrow (BM) in the **new Figure 1. L650-L652***

L 583. Please state where the sections were made, femur or tibia.

*Response: We apologized for the lack of clarity. The images are from the proximal tibia of the diseased knee joint. The respective information has now been added to the revised manuscript. **L651***

L586. n=5-6 per group as indicated per time point. Why is “as indicated” here?

Response: We thank for the critical comment. As the number of animals slightly varies, the individual values of all quantitative experiments were plotted as dots on the graphs. Thus, the corresponding group numbers are reported “as indicated” in the graphs.

Figure 2

A. Please state what the images are revealing for readers in the figure legend... meaning of pink and blue stain, articular cartilage ulceration etc.

*Response: We thank for the critical comment. This information has been added to the revised manuscript. **L664-L665***

Figure 3

Please provide pre surgery, WT, mutant representative site matched images for both 4 and 8 weeks.

*Response: We thank for the critical comment. Site-matched images have now been included in the **new Figure 3**.*

Figure 5

A. can you also show micro -CT images from 8 weeks post-surgery?

*Response: We thank for the valuable suggestion. Micro-CT images from both time points are now presented in the **new Figure 5**.*

Figure 6.

A. Please expand figure legend to state what you are showing in these histological images (for example for lower row hyperplasia and synovial hypertrophy are evident).

*Response: We thank the reviewer for the critical comment. The respective figure legend has been modified accordingly. **L714-L715***

References:

- 1 Glasson, S. S., Chambers, M. G., Van Den Berg, W. B. & Little, C. B. The OARSI histopathology initiative - recommendations for histological assessments of osteoarthritis in the mouse. *Osteoarthritis Cartilage* **18 Suppl 3**, S17-23, doi:10.1016/j.joca.2010.05.025 (2010).
- 2 Ferland, C. E., Laverty, S., Beaudry, F. & Vachon, P. Gait analysis and pain response of two rodent models of osteoarthritis. *Pharmacol Biochem Behav* **97**, 603-610, doi:10.1016/j.pbb.2010.11.003 (2011).
- 3 Ängeby Möller, K., Aulin, C., Baharpoor, A. & Svensson, C. I. Pain behaviour assessments by gait and weight bearing in surgically induced osteoarthritis and inflammatory arthritis. *Physiol Behav* **225**, 113079, doi:10.1016/j.physbeh.2020.113079 (2020).
- 4 Hoff, A. O. *et al.* Increased bone mass is an unexpected phenotype associated with deletion of the calcitonin gene. *The Journal of clinical investigation* **110**, 1849-1857, doi:10.1172/jci14218 (2002).
- 5 Schinke, T. *et al.* Decreased bone formation and osteopenia in mice lacking alpha-calcitonin gene-related peptide. *J Bone Miner Res* **19**, 2049-2056, doi:10.1359/jbmr.040915 (2004).

- 6 Inglis, J. J. *et al.* Regulation of pain sensitivity in experimental osteoarthritis by the endogenous peripheral opioid system. *Arthritis & Rheumatism* **58**, 3110-3119, doi:<https://doi.org/10.1002/art.23870> (2008).
- 7 Han, B. *et al.* Differentiated activities of decorin and biglycan in the progression of post-traumatic osteoarthritis. *Osteoarthritis Cartilage* **29**, 1181-1192, doi:10.1016/j.joca.2021.03.019 (2021).
- 8 Moilanen, L. J. *et al.* Monosodium iodoacetate-induced inflammation and joint pain are reduced in TRPA1 deficient mice--potential role of TRPA1 in osteoarthritis. *Osteoarthritis Cartilage* **23**, 2017-2026, doi:10.1016/j.joca.2015.09.008 (2015).
- 9 Renaudin, F. *et al.* NADPH oxidase 4 deficiency attenuates experimental osteoarthritis in mice. *RMD Open* **9**, doi:10.1136/rmdopen-2022-002856 (2023).
- 10 Seifer, P. *et al.* The Matrilin-3 T298M mutation predisposes for post-traumatic osteoarthritis in a knock-in mouse model. *Osteoarthritis Cartilage* **29**, 78-88, doi:10.1016/j.joca.2020.09.008 (2021).
- 11 Sophocleous, A., Börjesson, A. E., Salter, D. M. & Ralston, S. H. The type 2 cannabinoid receptor regulates susceptibility to osteoarthritis in mice. *Osteoarthritis Cartilage* **23**, 1586-1594, doi:10.1016/j.joca.2015.04.020 (2015).

Reviewers' comments:

Reviewer #1 (Remarks to the Author):

I thank the authors for the comments and the added research conducted. There is still one issue that needs to be addressed in the paper that I highlighted in the attached file.

It is in regards to the lack of pain measurements. The literature does not support the authors claims that the ACLT model does not show changes in pain behaviour and given the importance of nociception in OA, and aCGRP in nociception, this needs to be addressed.

Reviewer #2 (Remarks to the Author):

The reviewer thanks the authors for their efforts in preparing a rebuttal and thorough revision of their manuscript.

Authors have adequately addressed the queries.

Reviewer #3 (Remarks to the Author):

The investigators have comprehensively and satisfactorily addressed the concerns I raised. The results are an important contribution of new knowledge towards a better understanding of the complex pathophysiological mechanisms involved in joint organ degeneration in post traumatic osteoarthritis.

I have an additional few very minor suggestions.

L 661. The figure legend mentions a growth plate but there is none in this figure. In figure 2 please indicate that the arrows are pointing to cartilage loss or ulceration rather than simply degeneration.

R2 point-to-point reply for COMMSBIO-23-1803A

'The PCT/CT transcript of the CALCA gene controls cartilage degradation and subchondral bone loss, while α CGRP promotes osteophyte formation in murine post-traumatic osteoarthritis'

Reviewer #1 (Remarks to the Author):

I thank the authors for the comments and the added research conducted. There is still one issue that needs to be addressed in the paper that I highlighted in the attached file.

It is in regards to the lack of pain measurements. The literature does not support the authors claims that the ACLT model does not show changes in pain behaviour and given the importance of nociception in OA, and α CGRP in nociception, this needs to be addressed.

Response: We thank the reviewer for pointing out this important issue. Unfortunately, we are not able to perform the pain outcome measurements, as the method is not established in our laboratory and we lack a suitable collaborator with expertise in this area. However and as suggested by the editor, we have included this point as a major limitation of our study in the modified Discussion section. The relevant statement now reads as follows: "In this regard, the current study has several limitations. First, a major limitation is that we did not assess pain outcomes in the respective mouse lines, because these measurements are not routinely performed in our laboratory. As the ACLT model is associated with changes in pain behavior, and given the importance of nociception in ptOA and the role of α CGRP in nociception, future studies are warranted to address a corresponding influence on ptOA progression."

Reviewer #2 (Remarks to the Author):

The reviewer thanks the authors for their efforts in preparing a rebuttal and thorough revision of their manuscript.

Authors have adequately addressed the queries.

Response: We thank the reviewer for these positive remarks.

Reviewer #3 (Remarks to the Author):

The investigators have comprehensively and satisfactorily addressed the concerns I raised. The results are an important contribution of new knowledge towards a better understanding of the complex pathophysiological mechanisms involved in joint organ degeneration in post traumatic osteoarthritis.

I have an additional few very minor suggestions.

L 661. The figure legend mentions a growth plate but there is none in this figure. In figure 2 please indicate that the arrows are pointing to cartilage loss or ulceration rather than simply degeneration.

Response: Thank you to the reviewer for this comment. As suggested, we have changed the legend of Figure 2 to indicate that the arrows point to cartilage ulceration and have also removed the term "growth plate".